# The Role of BMI1 in Late-Onset Sporadic Alzheimer’s Disease

**DOI:** 10.3390/genes11070825

**Published:** 2020-07-21

**Authors:** Ryan Hogan, Anthony Flamier, Eleonora Nardini, Gilbert Bernier

**Affiliations:** 1Stem Cell and Developmental Biology Laboratory, Hôpital Maisonneuve-Rosemont, 5415 Boul. l’Assomption, Montreal, QC H1T 2M4, Canada; ryan.hogan@umontreal.ca; 2Whitehead Institute of Biomedical Research, 455 Main Street, Cambridge, MA 02142, USA; aflamier@wi.mit.edu (A.F.); enardini@wi.mit.edu (E.N.); 3Department of Neuroscience, University of Montreal, Montreal, QC H1T 2M4, Canada

**Keywords:** BMI1, Alzheimer’s disease, late-onset, sporadic, epigenetics

## Abstract

Late-onset sporadic Alzheimer’s disease (LOAD) seems to contain a “hidden” component that cannot be explained by classical Mendelian genetics, with advanced aging being the strongest risk factor. More surprisingly, whole genome sequencing analyses of early-onset sporadic Alzheimer’s disease cohorts also revealed that most patients do not present classical disease-associated variants or mutations. In this short review, we propose that *BMI1* is possibly epigenetically silenced in LOAD. Reduced *BMI1* expression is unique to LOAD compared to familial early-onset AD (EOAD) and other related neurodegenerative disorders; moreover, reduced expression of this single gene is sufficient to reproduce most LOAD pathologies in cellular and animal models. We also show the apparent amyloid and Tau-independent nature of this epigenetic alteration of *BMI1* expression. Lastly, examples of the mechanisms underlying epigenetic dysregulation of other LOAD-related genes are also illustrated.

## 1. Aging as the Number One Risk Factor of Alzheimer’s Disease

Among the neurodegenerative diseases, the etiology of Alzheimer’s disease (AD) has proven particularly difficult to sort out. Classically, dominant mutations in three genes, *Amyloid-beta precursor protein* (*APP*)*, Presenilin-1* (*PSEN1*) and *Presenilin-2* (*PSEN2*), are responsible for familial early-onset AD (EOAD) while sporadic late-onset AD (LOAD) can be associated with genetic risk factors like *APOE* and *CLU* in some, but not all, cases [1]. However, recent studies call into question our understanding of proposed Mendelian origins of this complex neuropathology. Highly penetrant mutations in *APP*, *PSEN1* and *PSEN2* are only present in a relatively small portion of sporadic early-onset AD cases [2]. Moreover, these known pathogenic gene variants are also found in some LOAD cases [3]. This suggests two, not mutually exclusive possibilities: the penetrance of these variants is not as high as previously thought; there are significant environmental or genetic factors mitigating the development and progression of the disorder. Recent genetic screenings have yielded dozens of novel possible mutations in over 20 candidate genes associated with sporadic early-onset AD cases [4]. However, a definitive list of AD-causing gene variants remains to be enumerated. Nonetheless, a primary risk factor of LOAD, and adult neurodegenerative diseases generally, has been determined—aging [5]. Between the ages of 80 to 84, the prevalence of LOAD is approximately 1 in 10 for women and slightly less for men; prevalence increases consistently and above the age of 95, it peaks at approximately 5 in 10 for women and 4 in 10 for men [5]. As aging is largely driven by the accumulation of DNA damage and epigenomic perturbations, see review in Maynard et al., 2015, a better understanding of the link between aging and LOAD is fundamental to our understanding of this disease [6]. 

## 2. BMI1 

### 2.1. B-Cell Specific Moloney Murine Leukemia Virus Integration Site 1

(*BMI1*, also called *PCGF4*) is involved in embryonic development, cell cycle regulation, DNA damage response, senescence, stemness and cancer [7,8]. *BMI1* encodes a 37 kDa protein composed of 326 amino acids with a highly conserved structure [7]. Regulation of *BMI1* expression includes numerous transcription factors and miRNAs, well summarized in Bhattacharya et al., 2015 [7]. Although several correlative relationships have been established, generally the mechanism of action is not well defined. The classical function of BMI1, as an integral part of the Polycomb Repressive Complex I (PRC1), is to maintain chromatin compaction both independent and dependent of histone H2A modification [9,10]. Specifically, PRC1 has E3 ligase activity that catalyzes monoubiquitination of histone H2A at lysine 119 (H2A^ub^), which translates into reduced transcription or silencing of target genes [9,10,11]. A principal target of BMI1 repressive activity is the well-known marker and proposed inducer of cellular senescence, the *INK4a* locus which encodes the tumor-suppressors p16^INK4a^ and p14^ARF^ (p19^ARF^ in mice) [12]. BMI1-mediated repression of this locus is essential to maintain neural stem cell self-renewal [13,14]. Additionally, ChIP-seq experiments have revealed that BMI1 is enriched at repetitive sequences (i.e., LINEs, SINEs and LTRs), particularly in regions of constitutive heterochromatin (labeled by the histone mark H3K9^me3^), and required for heterochromatin formation and maintenance [15,16]. Finally, BMI1 is recruited to sites of DNA breaks in order to promote DNA damage response (DDR) and repair [17,18]. Even upstream of the DDR, BMI1 is associated with mitigation of ROS buildup and the resulting oxidative damage, in part through inhibition of p53 pro-oxidant activity [19,20]. Not surprisingly, *Bmi1^−/−^* and *Bmi1^+/−^* mice also display several features of premature aging, especially in the central nervous system [15,19,20]. Based on these data, BMI1 seems to have a range of biological functions in the maintenance of both epigenetic and genetic integrity.

### 2.2. Reduced BMI1 Expression is Associated with Aging

In vitro, *BMI1* levels are reduced in pre-senescent (late passage) human and mouse fibroblasts compared to both early passage and quiescent controls, and overexpression of *BMI1* increases the replicative life span by four population doublings [21]. In vivo, the cortices of old mice (21−26 months old) show a substantial downregulation of *Bmi1* expression in terms of both mRNA (~60%) and protein (~40%) compared to young mice (two months old) [22]. Reduced *Bmi1* expression is unique to cortical neurons and is not observed in astrocytes [22]. Lastly, as expected, mRNA expression of *p16*
^*INK4a*^ and *p19^ARF^* were significantly increased in old mice [22]. In accordance with these data from mice, BMI1 immunoreactivity is notably reduced in human retinal neurons in old individuals compared to young individuals [22]. Despite these observations, *BMI1* downregulation has not yet been demonstrated to occur in the normal aging human brain. Importantly, the epigenetic mechanisms governing reduced *BMI1* expression during aging are largely unknown.

## 3. Neurodegenerative Hallmarks of Alzheimer’s Disease

### 3.1. Histopathology

Cardinal features that set AD apart from nonpathological aging include the presence of extracellular amyloid beta (Aβ) plaques, intracellular neurofibrillary tangles (NFT), synaptic atrophy and neurodegeneration (Figure 1) [23,24]. However, the question remains unresolved whether Aβ and NFT pathologies are upstream or downstream of other AD pathologies, including cognitive deficits [25,26].

For a plethora of reasons, well summarized in Wang et al., 2008, it is postulated that LOAD is driven by an epigenetic mechanism [27]. There are now numerous data, some of which find hypermethylation and hyper-hydroxymethylation in the LOAD-affected brain however other data show no significant differences in global methylation [28,29,30,31,32]. Ultimately, global genome methylation may not shed much light on the etiology of LOAD. Nonetheless such snapshots of genome methylation may be useful as an indicator of LOAD-risk [32]. It may prove more useful to examine methylation of specific genes, even individual single nucleotide polymorphisms (SNPs), in tissues of interest (i.e., the cortex and the hippocampus) to understand the origins of the disease. Examples of such investigations are discussed below. Meanwhile, zooming out from DNA methylation, LOAD-associated epigenome abnormalities also include reduced heterochromatin compaction in conjunction with smaller and more numerous chromocenters in cortical neurons [15,33]. 

In addition to epigenetic dysregulation in the case of LOAD, activated DDR elements suggest an increased level of DNA damage. Firstly, activated DDR proteins such as p-ATM, p-ATR and p-Chk1 are at higher levels in LOAD brains compared to control and EOAD samples [15]. These observations are also interesting considering that LOAD brains possibly exhibit a diminished repair capacity that is not limited to the regions most affected by AD pathologies [34]. It is important to note that DNA damage is a known inducer of senescence [35]. Likewise, expression of the *INK4a* locus has been associated with LOAD in numerous studies and markers of repression (i.e., H2A^ub^) were found to be depleted at the *INK4a* locus in LOAD but not in control brains [36,37,38]. 

### 3.2. BMI1 is Reduced in LOAD

It was found that *BMI1* is reduced at the mRNA and protein levels in LOAD brains when compared to those of age-matched, nondemented controls, suggesting epigenetic downregulation. Analysis of publicly available RNA-seq data of LOAD brain samples showed that *BMI1* reads were significantly lower in LOAD compared to controls, vascular dementia and mixed dementia samples. Interestingly, reads of the adjacent locus, *COMMD3*, and the fusion, *COMMD3-BMI1*, did not significantly differ between these groups [39]. The reduction of *BMI1* mRNA and protein was confirmed in the hippocampus and frontal cortex of LOAD samples [15,39]. Remarkably, in a comparison of LOAD, EOAD and age-matched controls, *BMI1* and H2A^ub^ reduction in the cortex were unique to LOAD [39]. Similarly, a mouse model of EOAD, *APP^swe^/MAPT^P301L^/PSEN1^M146V^* (3xTG mice), does not show changes in Bmi1 protein levels compared to Wild Type mice [39]. Furthermore, human neurons over-expressing the mutant *PSEN1* and *APP* genes do not present reduced *BMI1* expression [39]. These observations in EOAD brains, 3xTG mice and cultured neurons thus suggest that reduced *BMI1* expression cannot be a simple consequence of Aβ or Tau pathologies [39]. Lastly, analysis of induced pluripotent stem cell (iPSC)-derived neurons from four unrelated LOAD patients also revealed reduced *BMI1* mRNA and protein levels [39]. During the reprogramming of aged fibroblasts into iPSCs, important markers of aging are lost including heterochromatin markers, modification of the DNA methylation pattern, DNA damage and ROS levels [40,41]. Therefore, it is notable that despite the reprogramming required for iPSC generation from fibroblasts and the subsequent redifferentiation into neurons, *BMI1*-associated anomalies persist. This could suggest the existence of LOAD-associated genetic variant(s) affecting *BMI1* expression, incomplete epigenetic reprogramming at specific loci or the restoration of pathological epigenetic changes upon neuronal differentiation. 

## 4. Models of BMI1-Deficiency Recapitulate LOAD

Given the data outlined above that reduced *BMI1* expression is closely associated with aging and more importantly with LOAD, several models of *BMI1* deficiency have been developed and characterized to further explore this relationship. These include *BMI1* knock-down (KD) or knock-out (KO) using either shRNA or CRISPR-Cas9 in human iPSC-derived neurons, as well as *Bmi1^−/−^* and *Bmi1^+/−^* mouse models [13,14,15,16,19,39,42]. The main pathological features of LOAD discussed above are present in these models.

As cornerstones of LOAD histopathology, it is imperative that models of the disease replicate Tau and Aβ pathologies. For example, *BMI1* KD in 2D neuronal cultures induced p-Tau and Aβ accumulation along with neurodegeneration (Figure 1) [39]. Going further, *BMI1* KD in 3D neuronal cultures resulted in large extracellular amyloid deposits and intracellular p-Tau accumulations [37]. In a broader analysis of Tau and Tau-related proteins (e.g., GSK3β, p-GSK3β (Ser9) and p53), all were significantly increased in *BMI1*-deficient neuronal cultures; on the contrary, *BMI1* overexpression in naive neurons had the reverse effect on these proteins [39]. One general mechanism proposed to explain the accumulation of toxic proteins in AD is defective proteostasis (Figure 1) [43]. Notably, drug-mediated stabilization of p53 in naive neurons apparently impairs proteasome function and leads to significant p-Tau and Aβ accumulations when combined with GSK3β overexpression [39]. Conversely, dual inhibition of p53 and GSK3β largely rescues AD-related pathologies in *BMI1*-knockdown neurons. Moreover, BMI1 binds to *MAPT* (encoding for Tau) regulatory elements in both neuronal and human brain extracts and can repress *MAPT* transcription upon overexpression in cultured human neurons. Thus, combined inhibition of the p53, GSK3β and *MAPT* pathological triad may explain most of BMI1 protective activities against p-Tau and Aβ accumulation in human neurons. Lastly, in mouse models, increased levels of total Tau, p-Tau, Bace1 and amyloid plaques (although very rare) were shown in aged *Bmi1*-haplodeficient (*Bmi1^+/-^)* mice; these trends were present in the cortex and hippocampus, but absent in the cerebellum, showing specificity to LOAD-associated brain regions [15]. Notably, inhibition of ATM and ATR using a small molecule greatly mitigates the accumulation of p53 and p-Tau in *Bmi1*-deficient neurons, suggesting that genomic instability and constitutive activation of the DNA damage response machinery also contribute to the neurodegenerative process. Critical LOAD-associated neuropathologies also include axonal swelling, synapse loss and apoptosis [44]. Hence, *BMI1* KD and KO human neurons also present dystrophic neurites and axonal swelling [39]. Both in vitro and in vivo, *BMI1* deficiency leads to a significant reduction in PSD-95 and synaptophysin, indicative of synapse loss [15,39]. Finally, analysis of the cortex and hippocampus of *Bmi1*^+/−^ mice showed increased markers of apoptotic neurons [15,42]. 

Unsurprisingly given BMI1’s role as a repressor of the *INK4a* locus, discussed above, *Bmi1*-deficient murine models show clear signs of senescence both at the cellular and organismal level. In the haplodeficient model, markers of senescence, notably *p16^INK4a^* and *p19^ARF^* mRNA are significantly increased in cortical and hippocampal tissues [15,42]. Markers of neuronal cell death and stress, such as p53 and p-JNK, were also significantly increased [15]. Macroscopic and behavioral phenotypes associated with aging and LOAD are observed in *Bmi1*-deficient mouse models as well. For example, hair and weight loss, lens cataracts, pathological reflexes together with reduced hippocampal long-term potentiation (LTP) and behavioral deficit in spatial memory are reported in such models, whereas these pathologies are absent in age-matched Wild Type and young control groups [15,19,42]. Importantly, LTP is a cellular surrogate of memory, and short-term memory and spatial memory deficits are behavioral hallmarks of AD. 

In line with its function as part of PRC1, the ramifications of *BMI1* deficiency include de- repression of repetitive elements of the genome, in particular, pericentromeric repeats, and ALU, SINE and LINE sequences [16]. Hence, based on chromatin immuno-precipitation and chromatin fractionation assays, it is estimated that about 50% of the BMI1 protein pool is associated with genomic repeats. Other important markers of heterochromatin loss include significantly reduced levels of H3K9^me3^, H2A^ub^, HP1 and DEK1 [15,16]. Analysis of the nucleus reveals further abnormalities along these lines. Firstly, the chromocenters of *BMI1*-deficient cells and models were smaller and more numerous, suggesting de-nucleation of heterochromatin [15,16]. Secondly, nuclear morphology was altered, included nuclear envelope irregularities and piknotic nuclei—indicative of cell death. Importantly, these structural anomalies are prominent in LOAD neurons in situ when compared to age-matched controls (Figure 1) [16,20,33,43]. As in LOAD, *Bmi1* hemi-deficiency in mice results in significantly increased levels of DDR proteins such as p-ATM and p-ATR; moreover, the DDR proteins are found to be enriched at genomic repeats, suggesting that uncondensed heterochromatin may be more vulnerable to DNA damage and less efficient at repair (Figure 1) [39]. Mitochondrial dysfunction and a subsequent increase in ROS and markers of oxidative stress has been shown specifically in the mouse hippocampus, but also in various other cell types upon *Bmi1* deficiency [19,20,22,42]. At the mechanistic level, Bmi1 inhibits p53 accumulation in neurons, which otherwise binds to the promoter of antioxidant response genes to repress transcription. On the other hand, Bmi1 can also inhibit the transcription of key pro-oxidant genes. Conversely, *Bmi1* overexpression in primary mouse neurons reduces ROS levels, upregulates antioxidant defenses and greatly improves resistance to toxins such as camptothecin and 3-NP [22]. Bmi1 thus displays strong activity to protect cells against oxidative stress.

All together these data provide a well-rounded rationale that supports *BMI1* deficiency and haplo-insufficiency as models of LOAD. As a very reliable inducer of the LOAD state, it stands to reason that *BMI1* may be at the crux of the LOAD etiology in humans. While the mechanism underlying abnormal *BMI1* expression in LOAD remains a mystery (Figure 1), one possibility is that *BMI1* downregulation represents a cellular response against oncogenic transformation or DNA damage, leading to neuronal senescence [45]. Importantly as well, the timing of altered *BMI1* expression relative to the disease process is still an open question that needs to be answered.

## 5. Epigenetics

In an attempt to understand the mechanism of *BMI1* dysregulation in LOAD, we prioritize an epigenetic basis. Epigenetic regulation may entail one or several features such as DNA methylation, histone modifications, or post-translational means like noncoding RNAs. Given that epigenetic modifications are stable, yet reversible and can accrue over one’s lifetime, they represent a potentially fruitful field of investigation. Moreover, postmitotic cells like neurons are especially prone to accumulate epigenetic changes. For a more complete review of epigenetics and AD, see Qazi et al., 2018 [46]. In particular, a number of LOAD-associated genes have already been the subject of epigenetic studies and illustrate regulatory mechanisms that may apply to *BMI1* expression. These genes include *PSEN1, SIRT2* and *LRP6* and the epigenetic processes concerned are DNA methylation, SNP’s impact on miRNA binding and alternate splice variants, respectively. Lastly, an additional transcriptional regulator, REST, exhibits interesting similarities and differences compared to BMI1 in the context of LOAD.

### 5.1. PSEN1

Presenilin-1 plays an integral role in Aβ pathology as one of the core proteins of the γ-secretase complex. Noted above, highly penetrant *PSEN1* mutations are responsible for a significant number of familial, EOAD cases, however it is possible that dysregulation of this locus also participates in the etiology of sporadic, LOAD. In vitro models have shown that hypomethylation of the *PSEN1* promoter is associated with increased expression [47,48]. Such hypomethylation is not caused by amyloid production, in the case of a transgenic AD mouse model [49]. Similarly, a recent investigation of *PSEN1* methylation in human AD cortical samples reports a significant, inverse correlation between promoter methylation and gene expression [50]. This study examines both CpG and non-CpG methylation. Non-CpG methylation is especially interesting in the context of AD, as the authors suggest, because non-CpG methylation is largely the result of de novo methylation by DNMT3. DNMT3 not only has notable expression in the brain, its mechanism does not rely on cell replication which, of course, is absent in the post-mitotic neurons of the adult cortex [50]. 

### 5.2. SIRT2

The sirtuin family (*SIRT1-7*) has important functions in a diverse range of cellular processes. For a more complete review of the sirtuins in relation to AD, see Cacabelos et al., 2019 [51]. Here we focus on the *SIRT2* gene which contains multiple SNPs that have been associated with AD, albeit with very weak instances of increased risk [51,52,53,54]. However, what is notable is the case of two particular SNPs in the 3’UTR that are highly associated with AD risk. Mechanistically, the variants had significant effects on miRNA binding and therefore SIRT2 protein levels [54]. As analysis tools become more powerful and more efficient, it will be enlightening to study the interactions between multiple genetic risk factors. In this vein, a recent study measured the response of AD patients to a multifactorial treatment based on genotype. Metrics included a psychometric analysis and blood parameters which were measured before, during and after one year of treatment. Surprisingly, the results revealed a differential response to treatment based on one's bigenic genotype of a particular *SIRT2* SNP and the *APOE* variants *ε2*, *ε3* and *ε4* [51]. 

### 5.3. LRP6

*LRP6*, a gene encoding a coreceptor in the Wnt/β-catenin pathway, has been implicated in numerous AD-related pathways [55]. Investigations of epigenetic mechanisms of *LRP6* downregulation focused on *LRP6* SNPs and a splice variant that are associated with AD. Firstly, two SNPs, designated 14e and 18e (located in the 14th and 18th exon of *LRP6*, respectively) comprise a bi-allelic haplotype that is associated with AD in a combined sample including all *APOE* genotypes. Interestingly, this association becomes highly significant in AD groups of only *APOE ε4*-negative individuals. While 14e is a nonsynonymous SNP resulting in a five-fold decrease in LRP6 activation, 18e is a synonymous SNP with no significant effect on protein function [56]. Nonetheless, neither SNP results in decreased expression [56]. Alternatively, splice variants represent an additional mechanism of epigenetic control. In the case of *LRP6*, expression includes an isoform in which exon 3 is omitted (*LRP6Δ3*). This isoform results in reduced signaling and its levels are significantly increased in AD brains; however, to date there is no SNP or related genetic element identified that may explain this splicing variant [57].

### 5.4. REST

The restrictive element-1 silencing transcription factor (REST), or NRSF, is canonically known as a nuclear transcriptional repressor acting on a plethora of neuron-related genes [58]. Its expression is robust in neural progenitor cells, waning over the course of neuronal differentiation. REST exercises transcriptional repression via interaction and recruitment of histone deacetylases, histone methyltransferases and demethylases, and methyl-CpG binding proteins. On the other hand, REST is known to activate certain target genes by recruiting proteins like TET3, NSD3 and Brahma (Brm) [58]. 

Although the line of research linking REST and AD is only just beginning to be explored, there are exciting results. Firstly, REST has a protective role against oxidative stress and subsequent cell death in the aging brain [59]. *REST* mRNA levels are found to be significantly increased in the aging prefrontal cortex compared to young controls [59]. Epigenetic regulation is clearly at play, given that *REST* levels increase as one ages, but the mechanism has not yet been clearly elucidated. Importantly, while *REST* mRNA levels are unaffected in LOAD brains, REST protein levels and localization to the nucleus of neurons is reduced when compared to age-matched elderly controls, thus producing a partial “loss of function” through nuclear exclusion [59]. In a comprehensive follow-up study, the researchers employed iPSC-derived neuronal precursors from LOAD patients and normal controls in lieu of cortical brain samples [60]. While there was no significant difference in *REST* mRNA or protein levels, REST nuclear localization was consistently reduced by ~50% in LOAD vs. control neurons [60]. This marks interesting differences between *REST* and *BMI1* dysregulation in LOAD. Notably, *REST* levels correlate strongly with cognitive function, and even in individuals with advanced LOAD neuropathology but no measurable dementia, REST protein levels are high in the nucleus [59]. Studying mechanisms of *REST* regulation during normal and pathological brain aging thus warrants continued investigation. Whether *BMI1* and *REST* display genetic or biochemical interactions, and whether a hierarchy exists between those genes during LOAD initiation or progression are new questions open for experimentation. 

## 6. Conclusions

Although *BMI1* has been explored rigorously in the context of various cancers, we have only begun to investigate the link between *BMI1* and LOAD. Not only is *BMI1* expression reduced significantly in some aging tissues and senescent cells, but further reduction in the case of LOAD is unique, compared with other neurodegenerative disorders such as EOAD, frontotemporal dementia and Lewy body dementia. Likewise, *BMI1*-deficient models, both in vitro and in vivo, replicate the important hallmarks of LOAD. Given *BMI1* reduction at both mRNA and protein levels, epigenetic dysregulation seems likely however the mechanism remains elusive. Here several possibilities are illustrated by other LOAD-associated genes and gene variants. As the world’s aging population continues to grow, unraveling the mysteries of AD is essential for us to age with dignity. Neurodegenerative disorders have very few effective treatments at this time, so an appealing possibility is to delay the onset and mitigate the severity. Having such a large influence on our genetic landscape and, seemingly, an integral role in LOAD, further investigation into *BMI1* may be the key to creating future, therapeutic interventions. 

## Figures and Tables

**Figure 1 genes-11-00825-f001:**
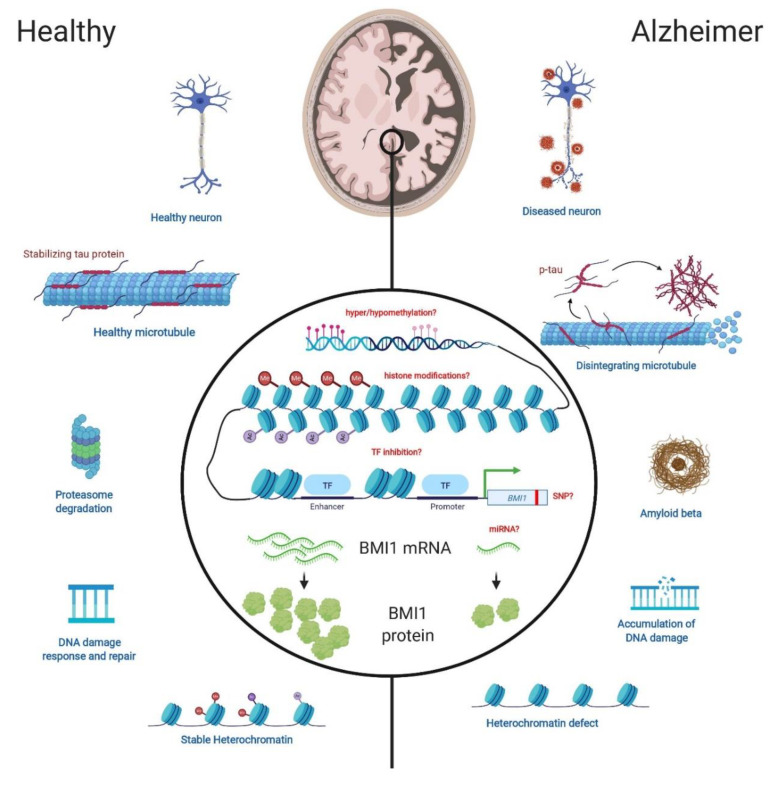
The role of BMI1 in LOAD-related neuronal pathologies and possible epigenetic regulatory mechanisms. Homeostatic processes in the healthy brain (left) versus the corresponding dysregulation that occurs in the LOAD brain (right). In models of *BMI1* deficiency, tauopathy and increased levels of AD-related proteins, such as GSK3β, are present. *BMI1* deficiency also results in Aβ accumulation which may be due, in part, to decreased proteasome activity. BMI1/RING1 induces p53 degradation in neurons and p53 accumulation inhibits proteasome activity. An increase in ROS and markers of oxidative stress in conjunction with higher levels of DDR proteins, indicative of higher levels of DNA damage, are prominent in the context of *BMI1* deficiency. Lastly, *BMI1* deficiency leads to heterochromatin de-nucleation and increases transcription of repetitive elements. Epigenetic regulation of *BMI1* has not yet been elucidated but may involve one, or more likely several, of the mechanisms listed as red text in the center of the image. TF: transcription factor, SNP: single nucleotide polymorphism, miRNA: microRNA.

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
