# Peer review of "The Role of BMI1 in Late-Onset Sporadic Alzheimer’s Disease"

_genes, 2020, doi:10.3390/genes11070825_

Round 1
Reviewer 1 Report
The review on BMI1 role in AD is sound and interesting. I have a couple of minor concerns that can be addressed by the authors:
1) The review is titled "epigenetics of BMI1" but no evidence for epigenetic regulation of this gene is provided. Or, at least, it remains too speculative. It is not clear to me whether the authors "suggest" that BMI1 is epigenetically modulated or they have evidence to provide. This point should be clarified and adequately addressed
2) The mechanistic relationship between BMI1 and AD is not clear. BMI1 appears to be directly involved in cell senescence and associated with Tau ad Abeta processing....but how this happens is not clear
3) Other genes showed epigenetic regulation in AD, particularly PSEN1 that has a direct effect on Abeta processig. This issue should be added to the "Epigenetics" paragraph
Author Response
1) The review is titled "epigenetics of BMI1" but no evidence for epigenetic regulation of this gene is provided. Or, at least, it remains too speculative. It is not clear to me whether the authors "suggest" that BMI1 is epigenetically modulated or they have evidence to provide. This point should be clarified and adequately addressed: We have changed the manuscript title and some sentences in the the main text accordingly, thank you.
2) The mechanistic relationship between BMI1 and AD is not clear. BMI1 appears to be directly involved in cell senescence and associated with Tau ad Abeta processing....but how this happens is not clear: This information was indeed missing in the document. We have corrected this by adding BMI1 molecular function(s) at repressing the p53, GSK3b and MAPT pathological triad. We have also added more information on BMI1 molecular function in the maintenance of genomic stability at heterochromatin and as an important modulator of ROS in neurons.
3) Other genes showed epigenetic regulation in AD, particularly PSEN1 that has a direct effect on Abeta processig. This issue should be added to the "Epigenetics" paragraph: This has been corrected accordingly, thank you.
Reviewer 2 Report
The short review by Hogan and colleagues deals with an important topic on how reduced BMI1 expression reproduces most LOAD pathologies in cellular and animal models. The topic is a timely and important addition to the literature, the manuscript is comprehensive and it may be of great interest to a wide scientific audience. I do recommend its publication after the following minor revision:
One can expect that this short review focuses in-depth on epigenetic modifications of BMI1 given that the title is “Epigenetics of BMI1 in Late-Onset Sporadic Alzheimer’s Disease” however epigenetic modifications in BMI1 are not the predominant part of this short review.
I suggest adding some sentences to contextualize the potential role of epigenetics of BMI1 in LTP, the cellular correlate of memory. Epigenetic modifications in BMI1 confined to specific neuronal ensembles might also be mentioned.
Another possibility is to change the title to “Role of BMI1 in Late-Onset Sporadic Alzheimer’s Disease” or similar.
Author Response
One can expect that this short review focuses in-depth on epigenetic modifications of BMI1 given that the title is “Epigenetics of BMI1 in Late-Onset Sporadic Alzheimer’s Disease” however epigenetic modifications in BMI1 are not the predominant part of this short review.: We have changed the manuscript title and the text in the main document accordingly, thank you.
I suggest adding some sentences to contextualize the potential role of epigenetics of BMI1 in LTP, the cellular correlate of memory.: LTP assays and spatial memory tests have been performed on Bmi1+/- mice. The mice indeed display major anomalies in LTP and spatial memory. This was mentioned in the previous document. Yet, we added the notion that LTP is the cellular correlate of memory, which is defective in AD patients (see herein). Thank you.
“Importantly, LTP is a cellular surrogate of memory, and short-term memory and spatial memory deficits are behavioral hallmarks of AD.”
Epigenetic modifications in BMI1 confined to specific neuronal ensembles might also be mentioned.
Another possibility is to change the title to “Role of BMI1 in Late-Onset Sporadic Alzheimer’s Disease” or similar.: We agree and have made the modification accordingly.